# The Importance of the Fatty Acid Transporter L-Carnitine in Non-Alcoholic Fatty Liver Disease (NAFLD)

**DOI:** 10.3390/nu12082178

**Published:** 2020-07-22

**Authors:** Dragana Savic, Leanne Hodson, Stefan Neubauer, Michael Pavlides

**Affiliations:** 1Radcliffe Department of Medicine, Oxford Centre for Magnetic Resonance Research, John Radcliffe Hospital, University of Oxford, Oxford OX3 9DU, UK; stefan.neubauer@cardiov.ox.ac.uk (S.N.); michael.pavlides@cardiov.ox.ac.uk (M.P.); 2Radcliffe Department of Medicine, Oxford Centre for Diabetes, Endocrinology & Metabolism, Churchill Hospital, University of Oxford, Oxford OX3 7LE, UK; leanne.hodson@ocdem.ox.ac.uk; 3Oxford NIHR Biomedical Research Centre, University of Oxford, Oxford OX3 7LE, UK; 4Translational Gastroenterology Unit, University of Oxford, Oxford OX3 9DU, UK

**Keywords:** L-carnitine, acetyl-carnitine, fatty liver, NAFLD, NASH, cirrhosis, fatty acid transport, Non-Alcoholic Fatty Liver Disease, Non-Alcoholic Steatohepatitis, liver health

## Abstract

L-carnitine transports fatty acids into the mitochondria for oxidation and also buffers excess acetyl-CoA away from the mitochondria. Thus, L-carnitine may play a key role in maintaining liver function, by its effect on lipid metabolism. The importance of L-carnitine in liver health is supported by the observation that patients with primary carnitine deficiency (PCD) can present with fatty liver disease, which could be due to low levels of intrahepatic and serum levels of L-carnitine. Furthermore, studies suggest that supplementation with L-carnitine may reduce liver fat and the liver enzymes alanine aminotransferase (ALT) and aspartate transaminase (AST) in patients with Non-Alcoholic Fatty Liver Disease (NAFLD). L-carnitine has also been shown to improve insulin sensitivity and elevate pyruvate dehydrogenase (PDH) flux. Studies that show reduced intrahepatic fat and reduced liver enzymes after L-carnitine supplementation suggest that L-carnitine might be a promising supplement to improve or delay the progression of NAFLD.

## 1. Introduction

One-third of the world’s Western population suffers from Non-Alcoholic Fatty Liver Disease (NAFLD) [1]. NAFLD develops when liver fat content exceeds 5% [2], and consists of a spectrum of pathologies ranging from simple steatosis (>5% liver fat) to Non-Alcoholic Steatohepatitis (NASH; fat + inflammation), fibrosis and cirrhosis. Liver fibrosis is a strong predictor of long-term mortality in patients with NAFLD [3,4]. Steatosis is associated with obesity and the metabolic syndrome [5] and develops when there is an imbalance between fatty acid uptake by the liver, synthesis (de novo lipogenesis) within the liver and disposal from the liver (Very Low Density Lipoprotein (VLDL) secretion and fatty acid oxidation). To enter the oxidation pathways fatty acids need to be coupled with L-carnitine in order to be transported into the mitochondria. Relative lack of L-carnitine may therefore lead to reduced fatty acid oxidation and triglyceride accumulation, resulting in NAFLD. Supplementation with L-carnitine may be a potential therapeutic option for lowering the risk of NAFLD by promoting fatty acid oxidation. Furthermore, studies show that L-carnitine is reduced in patients with liver disease, diabetes and cardiovascular disease [6,7,8] and supplementation with L-carnitine has been reported to improve liver inflammation and reduce liver enzymes and liver fat in these patients [9,10,11,12]. The aim of this review is to provide an overview of the published literature on the effects of L-carnitine supplementation in patients with NAFLD.

## 2. The Importance of L-Carnitine

L-carnitine (beta-hydroxy-gamma-N-trimethyl-aminobutyric acid) was discovered in 1905 as a constituent of muscle [13,14]. The name carnitine comes from the Latin word “*carnis*” (meat). The chemical structure was first described 20 years later, and it was in the 1970s that L-carnitine deficiencies were first discovered in humans [15].

The total body pool of L-carnitine is made up of several esters, including the short-chain ester, acetyl-carnitine [7]. L-carnitine homeostasis is regulated at multiple levels including intestinal absorption, de novo biosynthesis and renal reabsorption [16]. The human body contains approximately 300 mg/kg of L-carnitine, 98% of which is intracellular [17]. It is unevenly distributed between tissues and organs, with 80% present in muscle, and 5–10% present in the gastrointestinal tract. The liver contains about 3% and the blood contains 0.25% of the body’s L-carnitine pool [17].

### 2.1. Dietary Intake of L-carnitine

Dietary derived L-carnitine mostly comes from meat, therefore the Latin word “*carnis*”, with much smaller quantities from avocado and dairy products. In carnivores around 75% of the total body L-carnitine comes from the diet, but vegetarians mostly biosynthesize L-carnitine [18]. This is usually sufficient to maintain physiological function, as vegetarians also tend to have lower levels of long-chain fatty acids that need transportation to the mitochondria [19,20,21].

An average 70 kg person can absorb 23 to 135 mg/day of dietary L-carnitine, while a 70 kg person on a strictly vegetarian diet only takes up 1 mg/day of L-carnitine from the diet [22]. Infants are introduced to exogenous L-carnitine and acetyl-carnitine through breast milk, although it is known that L-carnitine and acetyl-carnitine have no specific immediate effects in the breast-feeding child [23]. It is speculated that L-carnitine in the breast milk serves as preparation for further uptake of L-carnitine through the diet, later in life.

For many years it was believed that all L-carnitine that was eaten through the diet was totally absorbed; however between 54–86% of L-carnitine from the food is absorbed, while only 5–25% of L-carnitine is absorbed if given through oral supplementation [24]. The efficiency of the absorption tends to diminish as the dose of L-carnitine increases, with some studies finding absorption of L-carnitine is saturated beyond doses of 2 g/12 h [25,26]. L-carnitine is absorbed partly through passive diffusion and partly via carrier-mediated transport in the colon and in the small intestines [22]. Dietary L-carnitine is degraded in the gut partly by trimethylamine N-oxide and partly by gamma-butyrobetaine [27]. The majority of these studies have investigated absorption in healthy humans and it is not clear whether there are any differences in L-carnitine absorption in individuals with NAFLD.

### 2.2. Endogenous L-carnitine Synthesis

L-carnitine is synthesized from the two amino-acids, lysine and methionine. A healthy human body can synthesize from 11 to 34 mg of L-carnitine per day. L-carnitine synthesis starts with the precursor trimethyl-lysine (TML), which is released from lysosomal protein degradation and ends with hydroxylation of gamma-butyrobetaine (BB) by gamma-butyrobetaine dioxygenase (BBD) producing L-carnitine (Figure 1). Trimethyl-lysine dioxygenase (TMLD) is the only enzyme involved in L-carnitine synthesis that is localized in the mitochondrial matrix. The other enzymes reside in the cytosol [28]. As the final enzyme in the synthesis pathway (BBD) is located in the liver, kidneys and in the brain, only these organs finalize the formation of L-carnitine from BB [29]. If the liver is metabolically compromised, then the last step of the synthesis pathway may be inhibited (Figure 1).

### 2.3. L-Carnitine Absorption

L-carnitine is transported into the cell by the high-affinity organic cation/carnitine transporter 2 (OCTN2), which mediates active absorption from the intestinal lumen into the enterocytes and the reabsorption into the kidneys. OCTN2 also mediates uptake into other tissues like adipose tissue, liver, cardiac myocytes, muscle cells, lymphocytes, and the brain.

### 2.4. Transport of Fatty Acids into Mitochondria

Within cells, long-chain fatty acids are dependent on esterification with L-carnitine to form acetyl-carnitine in order to be transported from the cytoplasm to the mitochondrial matrix for oxidation and energy production (Figure 2). The enzymes carnitine-palmitoyl-transferase-1 (CPT I), CPT II and carnitine-acylcarnitine translocase (CACT) are essential in catalyzing these reactions (Figure 2). CPT I allows binding of acyl-CoA to L-carnitine to form acyl-carnitine for entry into the mitochondrial intermembrane space. From there, CACT transports the acyl-carnitine across to the mitochondrial matrix in exchange for free L-carnitine. Once acyl-carnitine is inside the mitochondrial matrix CPT II separates it into L-carnitine and acyl-CoA. The free L-carnitine can then be exported to the cytosol by CACT. There is evidence that the composition of dietary fatty acids affects the CACT, by increasing the transcriptional rate of CACT RNA, and therefore fatty acid oxidation, in the mitochondria of livers in rats [30]. Within the liver, acyl-CoA then undergoes β-oxidation to produce acetyl-CoA for oxidative phosphorylation or ketone body production [31].

### 2.5. L-Carnitine as a Buffer for Excess Acetyl-CoA

In a reaction catalyzed by CPT I, the free form of L-carnitine binds to a fatty acid to form acyl-carnitine. Acyl-carnitines have varying chain-lengths, depending on the cellular location and metabolic purpose. Even though L-carnitine binds to all chain-lengths of fatty acids, it is only the most abundant form of long-chain fatty acids that are dependent on L-carnitine for transportation to the mitochondria [32]. However, intramitochondrial L-carnitine can export both short-chain and medium-chain acyl-CoA out of the mitochondria, that otherwise could lead to the production of free radicals which can destabilize the cell membrane; thus a reduced level of L-carnitine may lead to oxidative damage [33,34]. Both the tissue and the plasma will have a significant free and bound pool of L-carnitine. L-carnitine can be re-circulated once fatty acids enter the mitochondrial matrix, therefore only a small amount of L-carnitine is required in order to allow fatty acid to enter the mitochondria. However, L-carnitine has another important function as a buffer of excess acetyl-CoA in the mitochondria, through the formation of acetyl-carnitine [35]. A larger amount of L-carnitine is needed for this function, since acetyl-CoA either needs to be metabolized by the TCA cycle or exported with the use of L-carnitine in form of acetyl-carnitine. Once acetyl-carnitine is transported it can either be excreted in the urine or split from L-carnitine, which can be re-circulated again. L-carnitine reserves can be used up completely if there is a larger amount of excess acetyl-CoA that needs to be buffered out of the mitochondria [15].

## 3. Fatty Liver Disease and the Role of L-Carnitine

### 3.1. Drivers of Non-Alcoholic Fatty Liver Disease (NAFLD)

NAFLD is associated with obesity, insulin resistance [36,37], diabetes and cardiovascular disease (CVD) [38,39]. The main drivers in NAFLD are inflammation and accumulation of lipid [40]. Individuals with NAFLD have an intrahepatic accumulation of several lipid species, including diacylglycerides, triglycerides, ceramides and cholesterols [41]. Kupffer cells in the liver respond to alterations in lipid accumulation and can activate inflammatory pathways [42], that then can drive NASH progression. L-carnitine could be relevant to NAFLD pathology in two ways. Firstly, reduced levels of L-carnitine may lower fatty acid oxidation and be a contributing factor in the accumulation of liver fat. L-carnitine has been shown to have anti-inflammatory effects by upregulating the Peroxisome Proliferator Activator Receptor-γ (PPAR-γ) in the liver [43].

Other circumstantial evidence for the importance of L-carnitine in NAFLD comes from studies that have examined L-carnitine in associated metabolic conditions and that have shown that L-carnitine is reduced in obesity, insulin resistance, diabetes and advanced age [6,16,44,45]. Patients with obesity may present with reduced CPT I levels leading to reduced levels of intracellular L-carnitine, which prevents them from using certain fatty acids for energy production, resulting in lipid accumulation in the cells especially in the adipose tissue and the liver [46,47].

### 3.2. Fatty Liver Disease Is a Feature of Primary Carnitine Deficiency

Primary Carnitine Deficiency (PCD) is an autosomal recessive disorder of fatty acid oxidation due to the lack of organic cation/carnitine transporter (OCTN). The lack of OCTN, which is needed for the absorption of L-carnitine results in L-carnitine deficiency. Patients with PCD have low serum levels of L-carnitine and low intracellular levels of L-carnitine, thus fatty acids are not utilized as an energy source and accumulate [48], whilst patients rely exclusively on glucose for energy metabolism. As a consequence glucose stores are depleted rapidly and hypoglycemia is often seen in patients with PCD [49,50,51]. The inability to oxidize fatty acids also leads to an elevated production of reactive oxygen species (ROS) [52]. Half the patients with this condition have hepatomegaly and elevated transaminase and this condition is therefore an exemplar of the link between L-carnitine and liver health.

Patients with PCD have similarities in their liver profile to patients with NAFLD [51,53]. Patients with PCD show accumulation of fat in their liver and often they show hepatic encephalopathy, which is one of the major complications of advanced liver disease [50,54]. Furthermore in both diseases there are elevated liver enzymes like alanine transaminase (ALT) and aspartate transaminase (AST)—an indicator of liver injury [55]. A few cases in small children with PCD showed that liver size and enzymes were normalized after treatment with L-carnitine [52,56,57]

### 3.3. Other Evidence Linking L-Carnitine Deficiency to Liver Disease

Further evidence that lack of L-carnitine or dysfunction in the L-carnitine shuttle leads to a fatty liver comes from the effects of the drug etomoxir. Etomoxir is an irreversible inhibitor of CPT I and therefore switches energy metabolism from fatty acid to glucose oxidation in humans [58]. Etomoxir leads to elevated food intake and reduces liver energy status by reducing adenosine triphosphate/adenosine diphosphate (ATP/ADP) levels in rats [59]. Furthermore, in mice that were fed a short-term high fat (45%) diet, etomoxir stimulated glucose oxidation, peripheral glucose disposal, and led to elevated circulating fatty acids and circulating triglycerides (TGs) within 5 h of treatment and after several days hepatic steatosis was induced [60]. Another study that used etomoxir showed that inhibition of CPT I activity by 50% specifically in the liver led to an enlarged liver in a murine model [61].

These studies suggest that inhibition of the L-carnitine shuttle by pharmacological inhibition of CPT I, as in the case of etomoxir, results in liver steatosis. Reduced levels of L-carnitine would be expected to have a similar effect as they could equally result in an inability to transport fatty acids into the mitochondria for oxidation.

### 3.4. Patients with Chronic Liver Disease have Low Levels of L-Carnitine

There are several pieces of evidence pointing to low levels of L-carnitine in patients with chronic liver disease, particularly those with cirrhosis. This is critical since the liver is the primary site for L-carnitine synthesis and therefore impaired L-carnitine synthesis due to liver disease can result in whole-body impairment of L-carnitine metabolism [62]. It is unlikely that a reduction in L-carnitine causes NAFLD but having reduced levels of L-carnitine could exacerbate liver steatosis, and contribute to overall disease progression. The accumulation of fat can lead to a dysfunction in the biosynthesis of L-carnitine (Figure 1) in the liver thus inducing a negative feedback loop, whereby less L-carnitine is produced which leads to enhanced impairment of fatty acid oxidation re-enforcing fatty liver disease.

In a study of 68 children with chronic liver disease, 38 of whom had liver cirrhosis, serum L-carnitine levels were significantly lower in those with cirrhosis compared to those with early liver disease and healthy controls [62]. A correlation between TG levels and L-carnitine concentration in the plasma was found in patients with chronic liver disease, although this relationship was not detected in the patients with cirrhosis possibly due to already too low and sometimes absent levels of L-carnitine [62]. This could indicate that early liver disease progression might be evaluated not just based on fat levels in the liver but also on reduced levels of L-carnitine; however, once the stage of liver disease is severe, there is a complete depletion of L-carnitine and it is no longer of value.

Rudment et al. found that hospitalized patients with cirrhosis had significantly reduced levels of free and total L-carnitine in the serum compared to healthy controls [63]. In the same study, post-mortem examination of patients with cirrhosis that died also found reduced tissue L-carnitine levels in heart, liver, kidney, brain and muscle [63].

### 3.5. Acyl-Carnitine Chain Length can be Associated with Liver Disease

A study in 241 patients with biopsy-proven NAFLD and 23 patients with hepatocellular carcinoma (HCC) showed an inverse relation between disease severity and acyl-carnitine length. Plasma long-chain acyl-carnitine, but not free L-carnitine, was associated with fibrosis, inflammation and HCC. Medium-chain acyl-carnitines were reduced with worsening severity of NAFLD [64]. This inverse relationship is also supported by previous studies that have shown that replacement of dietary long-chain fatty acids with medium-chain fatty acids reduces triglyceride accumulation by 50% in the murine liver [65].

Another study in patients with cirrhosis also showed up to a three-fold elevation of short-chain acyl-carnitine and long-chain acyl-carnitine in the plasma compared to healthy volunteers [66], and this has been supported by others [62,67]. These studies suggest that the length of the acyl-carnitine might be critical for predicting the severity of patients with NAFLD; however, larger studies are needed to confirm these findings.

In urinary metabolomics Dong et al. showed that, compared to healthy controls, patients with NAFLD and NASH had reduced levels of urine free L-carnitine but not acetyl-carnitine [33]. This suggests that the body is conserving free L-carnitine to be able to bind to long-chain fatty acids, but it also shows an elevated excretion of short-chain acetyl-CoA. This study did not measure other lengths of acyl-carnitine.

## 4. The Importance of L-Carnitine Supplementation

L-carnitine supplementation has beneficial effects in patients with fatty liver disease, where elevations in high-density lipoprotein (HDL) cholesterol and reductions in liver fat have been reported [68]. L-carnitine has been shown to elevate activity and transcription of hepatic CPT I [69], leading to a reduced amount of fat in the liver [70]. Several studies have shown improvement in hepatic steatosis and cirrhosis after L-carnitine supplementation [11,50,71,72]. Furthermore, L-carnitine supplementation in humans and animal models has been shown to modulate insulin sensitivity and glucose uptake and also to have an antioxidant effect in hepatocytes [71,73,74].

### 4.1. L-Carnitine Supplementation is Beneficial to the Liver

Several studies have examined L-carnitine’s ability to reduce fat accumulation in the liver in patients with NAFLD, generally with positive results (Table 1). These studies show that liver enzymes that are most commonly used as a laboratory test for detecting an abnormal liver can be normalized with supplementation of L-carnitine [75].

In one animal study, plasma L-carnitine was reduced in obese animals that were fed a high-fat diet for a full year, and this was associated with reduced expression of hepatic regulatory L-carnitine genes. These effects were reversible with L-carnitine treatment [16]. We have also previously shown that streptozotocin (STZ)-induced diabetic rats treated with L-carnitine (3 g/kg/day for five weeks) showed improved liver enzymes as well as choline levels in the liver while reducing TGs in the plasma [76].

Just as L-carnitine induces improvements in NASH [77], it also may induce regression of cirrhosis. Patients with cirrhosis have an acquired L-carnitine deficiency [12] and levels of free fatty acids progressively elevate with the severity of liver cirrhosis [78].

There are some discrepancies in findings, and it is not entirely understood when L-carnitine elevates free fatty acids [81] and when L-carnitine reduces free fatty acid [82]. For example in patients with obesity, L-carnitine directly reduces all free fatty acids in the plasma by transporting them to the mitochondria [83]. One study investigated 13 patients with liver cirrhosis before and after four weeks of L-carnitine treatment (1800 mg/day), and found that, after L-carnitine treatment, free fatty acid levels increased, whole-body carbohydrate oxidation increased, whilst whole-body fatty acid and protein oxidation significantly decreased [81]. Others have reported no differences in energy metabolism when patients with liver cirrhosis were treated with L-carnitine, although improvements in exercise tolerance were observed, which alludes to improved energy metabolism [84]. Data from animal models of diabetes point towards an effect of L-carnitine on energy metabolism with one study showing that L-carnitine supplementation elevates fatty acid oxidation [85].

Intestinal microbiota of trimethylamine (TMA) is metabolized to trimethylamine-N-oxide (TMAO), which is associated with CVD risk and could promote atherosclerosis [86]. Koeth et al. found that L-carnitine supplementation in mice enhanced synthesis of TMA and TMAO. Unlike other studies included in this review, where patients were given oral treatments of pure L-carnitine, Koeth et al. investigated the production of TMAO in the plasma and the urine by feeding patients an eight-ounce steak followed by a capsule of 250 mg heavy isotope-labelled L-carnitine [86]. The increase in TMAO was modest as the authors suggest but in five subjects TMAO production could be suppressed by giving oral broad-spectrum antibiotics, suggesting how intestinal microbiota contribute to the link between red meat consumption and CVD risk [86]. Since these patients were given meat, the dietary uptake of L-carnitine would have been much higher compared to if the same dose of oral L-carnitine was given [24], possibly leading to higher TMAO levels. The reason why a larger consumption of L-carnitine occurs if taken through the diet, rather than as a supplement, is still unclear and need to be further investigated.

### 4.2. Effects of L-Carnitine in Ketogenesis

Studies have shown how L-carnitine stimulates ketogenesis in the liver of mice [87,88]. A study undertaken in perfused livers showed that ketogenesis increases with an infusion of free L-carnitine in the liver. A subsequent infusion of long-chain fatty acids had no effect on ketone production, which implies that the substrate of ketone production in the presence of L-carnitine is endogenous fatty acids [87]. Nakajima et al. showed the rate-limiting step in ketogenesis is L-carnitine and not fatty acid supply [87]. Reduced L-carnitine levels in the liver result in an impairment of ketogenesis after a fatty diet (80%) in children with primary carnitine deficiencies and episodes of hepatic and cerebral dysfunction [89]. Oral L-carnitine supplementation improved these children’s clinical outcome and restored L-carnitine levels in the plasma [89]. Several studies have also shown that L-carnitine lowers the ketone body β-hydroxybutyrate [24,88,90] and that it is dose dependent [91]. It is not yet clear how L-carnitine regulates ketone utilizations or production, but these studies would suggest that L-carnitine can activate different pathways [90,92,93].

### 4.3. L-Carnitine has Significant Effect on Insulin and Glucose Levels

Insulin resistance is often observed in individuals with liver disease, specifically in patients with NASH [94]. Patients with NAFLD have been shown to have an impaired ability to oxidize glucose and non-oxidative glucose disposal with insulin stimulation [95].

L-carnitine does not only affect fatty acids, but it also showed a significant effect on glucose and insulin levels in several studies [8,96]. Randle et al. proposed in 1963 that the interaction between carbohydrate and fatty acid metabolism takes the form of a glucose-fatty-acid cycle that controls blood glucose levels, fatty acid concentrations and insulin sensitivity [97,98]. The Randle cycle essentially describes that when fatty acid oxidation is active there is a significant reduction in the uptake and utilization of glucose by inhibition of PDH flux, but it also works the other way around, so the production of acetyl-CoA from glycolysis will inhibit fatty acid oxidation through the generation of malonyl-CoA which can inhibit CPT I [99].

A study in 25 healthy subjects evaluated the effects of L-carnitine on glucose metabolism where subjects were initially infused with 5% glucose solution and 48 h later the same subjects were infused with 2 g of L-carnitine with 5% of glucose solution [100]. It was found that L-carnitine reduced glucose levels by reducing insulin levels; notably, glucose was maintained in the normal range [100]. Furthermore, Bae et al. showed that patients with NAFLD and diabetes improved their glycemic control (using HbA_1c_ as a marker) after treatment with carnitine-orotate for 12 weeks [11].

There are several proposed mechanisms for L-carnitine’s effect on carbohydrate metabolism including: (i) the regulation of the ratio of acetyl-CoA/CoA in the mitochondria and thereby PDH flux [101], (ii) by modulating glycolytic and gluconeogenic enzymes [99,102], and (iii) by stimulating insulin-like growth factor-1 (IGF-1) signaling cascades [103]. Infusion of L-carnitine in healthy people attenuates a rise in plasma glucose levels with a 10% glucose infusion for 3 h [104]. Furthermore, insulin can stimulate free L-carnitine as measured in the muscle [105]. Insulin sensitivity was reported to be improved by inducing hepatic autophagy through PPAR-γ, a possible new mechanism of L-carnitine treatment [106].

Recently it was reported that inflammation can affect insulin sensitivity [107]; L-carnitine has been shown to improve insulin-stimulated disposal of glucose [108], therefore it is plausible that L-carnitine may play a role in reducing inflammation via improvements in insulin-stimulated disposal of glucose.

Despite the above findings, treatment with L-carnitine does not always show changes in insulin levels even though the majority of studies report reduced serum glucose levels, TGs, fatty acids and ketone levels [8,109], sometimes without insulin changes. The discrepancy remains unclear in findings of studies in insulin levels and the mechanisms underpinning the effect of L-carnitine on plasma insulin remain to be elucidated.

Associations have been shown between insulin resistance and elevated long-chain acyl-carnitines in the plasma [110,111], and patients with NAFLD have higher levels of long-chain acyl-carnitines [111]. Progression of NAFLD has been shown to correlate with long-chain acyl-carnitine species [64,111] with patients with cirrhosis also often have elevated plasma insulin levels. Mihalik et al. showed that insulin infusion could reduce all species of plasma acyl-carnitine in healthy people, but in patients with diabetes this function was blunted [112]. Plasma long-chain acyl-carnitine species are elevated in patients with NAFLD, obesity and type 2 diabetes, suggesting that more fatty acids can enter the mitochondria, but since there is also an elevation of short-chain acyl-carnitines there might be a defect in the oxidation pathways.

## 5. Summary and Conclusions

L-carnitine is a critical co-factor for transporting long-chain fatty acids into mitochondria for β-oxidation and to export excess acetyl-CoA from the mitochondrial matrix. It is relevant to liver disease in two ways. Firstly, the liver is critical in synthesizing L-carnitine and, if diseased, L-carnitine biosynthesis is reduced which may affect whole body fatty acid metabolism. Secondly, it might be a potential treatment for liver fat accumulation as it promotes fat oxidation and can also have beneficial effects on carbohydrate metabolism.

Treatment with L-carnitine can improve outcomes in patients with fatty liver disease (Figure 3) and has been shown to reduce ALT and AST levels, as well as liver fat accumulation. L-carnitine administration has also been shown to improve markers of glycemic control in patients with NAFLD and diabetes, most likely by regulating the ratio of acetyl-CoA/CoA in the mitochondria and thereby the PDH flux.

Patients with chronic liver disease often have reduced levels of L-carnitine. The length of acyl-carnitine species found in plasma and urine might be of importance for disease outcomes. Therefore, it might not be sufficient to solely investigate free L-carnitine, but rather all L-carnitine species should be studied. Several studies show elevated plasma long-chain acyl-carnitine but not free L-carnitine and medium-chain acyl-carnitine to be associated with fibrosis, inflammation and cirrhosis in patients.

Larger and more comprehensive studies are needed to confirm whether L-carnitine has the beneficial effects observed in these small-scale studies in patients with NAFLD. Studies generally only measure free L-carnitine, acetyl-carnitine, or both. If in vivo imaging or biopsies are available, a more comprehensive analysis of the L-carnitine species could be possible before and after treatment with L-carnitine, to fully understand the effects of L-carnitine supplementation. Measuring L-carnitine species in a range of phenotypes might provide an insight into their use as a biomarker for liver disease.

## Figures and Tables

**Figure 1 nutrients-12-02178-f001:**
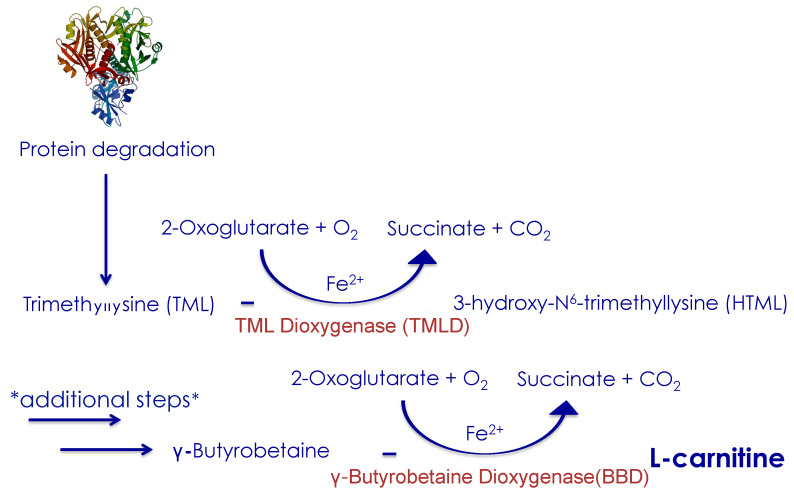
L-carnitine Synthesis. Endogenous L-carnitine synthesis. L-carnitine is biosynthesized from trimethyl-lysine (TML). At least four enzymes are involved in the overall biosynthesis pathway. These are trimethyl-lysine dioxygenase (TMLD), 3-hydroxy-N-trimethyllysine aldolase (HTMLA), 4-N-trimethylaminobutyraldehyde dehydrogenase (TMABA-DH) and γ-butyrobetaine dioxygenase (BBD). * The double arrow represents additional steps between HTML and γ-butyrobetaine, which are 3-hydroxy-N-trimethyllysine aldolase (HTMLA) that catalyzes 4-N-trimethylaminobutyraldehyde, which then is catalyzed by 4-N-trimethylaminobutyraldehyde dehydrogenase (TMABA-DH) into γ-butyrobetaine.

**Figure 2 nutrients-12-02178-f002:**
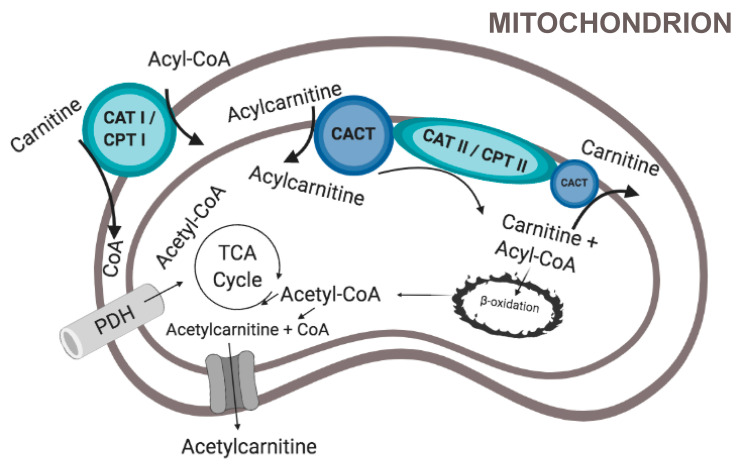
The L-carnitine shuttle. L-carnitine binds to acyl-CoA to help transportation to the mitochondria for β-oxidation. L-carnitine also binds to excess acetyl-CoA to be exported from the mitochondria. The outer mitochondrial membrane contains Carnitine Acyltransferase I/Carnitine Palmitoyl Transferase (CAT I/CPT I), that binds L-carnitine to acyl-CoA. Acyl-carnitine can then enter the intermembrane space of the mitochondria. The inner mitochondrial membrane contains Carnitine Acyl Carnitine Translocase (CACT), which both can transport acyl-carnitine into the mitochondrial matrix, and can export L-carnitine. The inter mitochondrial membrane also contains Carnitine Acyl Transferase II/Carnitine Palmitoyl Transferase II (CAT II/CPT II), which can separate L-carnitine from the acyl-CoA, so that it can undergo β-oxidation. L-carnitine can also bind to acetyl-CoA forming acetyl-carnitine in the mitochondrial matrix, which allows for export of acetyl-CoA, if not used for oxidative phosphorylation or ketone production. Figure created in BioRender.com.

**Figure 3 nutrients-12-02178-f003:**
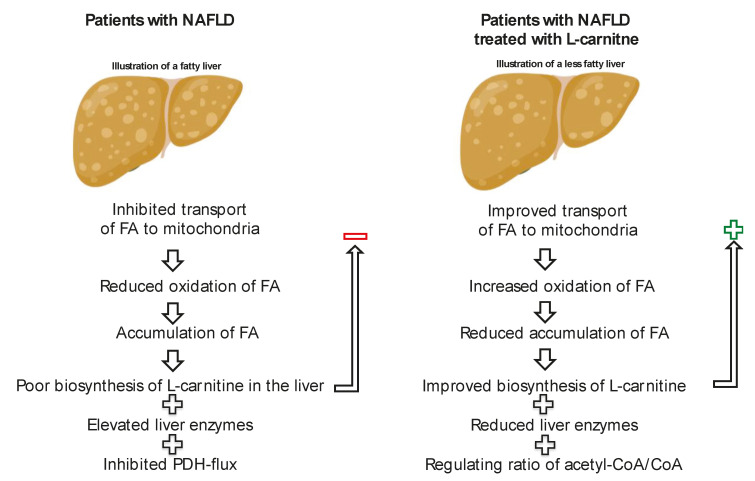
Simplified schematic of the processes in Non-Alcoholic Fatty Liver Disease (NAFLD) with and without L-carnitine treatment. Left is an illustration of a fatty liver, where there is reduced oxidation of fatty acids (FA) [113], which results in accumulation of FA [114], that can lead to a less functioning liver, with elevated liver enzymes [115], possibly resulting in poor biosynthesis of L-carnitine in the liver [29], which has a negative feedback loop on transport of FA into the mitochondria. Furthermore patients with NAFLD have inhibited pyruvate dehydrogenase (PDH) flux [116]. Right is an illustration of a fatty liver treated with L-carnitine. L-carnitine treatment will improve the transport of long-chain FA to the mitochondria, which will increase the oxidation of FA and therefore reduce accumulation of fat in the liver [11,71]. By having less fat in the liver, the biosynthesis of L-carnitine will not be inhibited by the liver, and it will have a positive feedback-loop and improve transport of FA to the mitochondria by L-carnitine. L-carnitine will therefore reduce liver enzymes and improve liver function [11]. L-carnitine treatment stimulates PDH flux by improving the acetyl-CoA/CoA ratio [101].

**Table 1 nutrients-12-02178-t001:** Research studies in humans. Effects of L-carnitine supplementation in liver disease. Most results are undertaken in blood samples unless otherwise stated. ALT = alanine aminotransferase, AST = aspartate transaminase, BW = body weight, CRP = C-Reactive Protein, CT = Computed Tomography, FA = fatty acids, HbA_1c_ = glycated hemoglobin, TNF = Tumor Necrosis Factor, LDL = low-density lipoprotein.

Author	Type of Study	Patients	Duration	Dose	Disease	Results
Alavinejadet al. [79]	Randomized double blind pilot study	60	3 months	750 mg of L-carnitine tablets	NAFLD + Diabetes	Reduced ALT levels
Somi et al. [71]	Placebo controlled trial	80	24 weeks	500 mg of L-carnitine twice/day	NAFLD	Reduced ALT, AST, BW Improved sonographic grade
Lim et al. [80]	Pilot Study	45	3 months	600 mg/day of L-carnitine	NAFLD	Reduced ALT, AST, bilirubin. Elevated peripheral mitochondrial copy number
Bae et al. [11]	Randomized control trial	78	12 weeks	150 mg of carnitine-orotate	NAFLD + diabetes	Reduced liver attenuation index on CT Reduced ALT, AST and HbA_1c_ levels
Malaguarna et al. [10]	Randomized double-blind placebo-controlled trial	74	24 weeks	1 g of oral L-carnitine twice/day	NASH	Reduced liver inflammation (biopsies) Reduced serum TNF-alpha & CRP Reduced insulin, LDL, cholesterol, TG.
Cecere et al. [9]	Randomized controlled clinical trial	31	4 weeks	6 g/day of L-carnitine	Cirrhosis	Reduced ammonia levels, improved psychometric test. Improved neurological function
Sakai et al. [81]	Pilot Study	13	4 weeks	1800 mg/day		Reduced Free FA Elevated whole-body carbohydrate oxidation. Decreased FA and protein oxidation

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
