# Peer review of "The Importance of the Fatty Acid Transporter L-Carnitine in Non-Alcoholic Fatty Liver Disease (NAFLD)"

_nutrients, 2020, doi:10.3390/nu12082178_

Round 1

Reviewer 1 Report

The manuscript entitled “The importance of the fatty acid transporter L-carnitine, in non-alcoholic fatty liver disease (NAFLD)” by Dragana Savic et al. describes an important role of L-carnitine for liver health, suggesting that L-carnitine supplementation could act as potential treatment for fat accumulation in the liver, promoting mitochondrial β-oxidation.

The manuscript has relevant information on the NAFLD findings and its correlation with L-carnitine. The scientific review is generally well written. Some improvements to the manuscript suggested below.

Line22: pyruvate dehydrogenase

Line 61: Include the reference

Line 80, 83, 101, 104 and 213: Put the corresponding references, since it mentions that you have an error.

Figure 1: Changing the word in the diagram “Deoxygenase” to “Dioxygenase”

Line 88 and 90: Changing the word “hydroxylase” to “Dioxygenase”

Line 121: Remove characters in the word “β-oxidation”

Line 128, 198 and 341: Changing the word TAGS to TAGs

Line 156: “an d” should be “and”

Line 172: Change the word l-carnitine” to “L-carnitine

Line 231: “hepatcolellular” should be “hepatocellular” 

Line 334: “Furtermore” should be “Furthermore”

Line 50: I recommend removing the subtitle “2. Materials and Methods”, therefore it would remain “2. The importance of L-carnitine, 3. Fatty Liver Disease and the Role of L-carnitine, 4. The importance of L-carnitine Supplementation” and “5. Summary and Conclusion”.

Check the table since the words have moved.

To complete the review, I suggest adding a scheme showing its possible effect of L-carnitine on NAFLD. This would further enhance this scientific review.

Author Response

Dear Reviewer

Thank you for your feedback and for making this manuscript even better. We have addressed all the comments as seen below. The lines refer to the line in ‘tracked changes’ mode.

  • Line 22: edited to pyruvate dehydrogenase
  • Line 173: The reference [18] has been included.
  • Line 204, 208, 333, 335, 886: We have added the reference to “figure 1” and “figure 2”.
  • Figure 1: The word in the diagram has been changed to “Dioxygenase”
  • Line 319 and 321: The word hydroxylase has been changed to “Dioxygenase” in the figure text.
  • Line 426: Characters in the word “beta-oxidation” have been removed
  • Line 1129, 1455 the word TAGS have been changed to TAGs
  • Line 550: Has been changed to “and”, and paragraph has been re-written
  • Line 559: l-caritine has been edited to “L-carnitine”
  • Line 902: “hepatcolellular” has been edited to “hepatocellular”
  • Line 1447: “furthermore” has been edited to “Furthermore”
  • Line 165: The heading “Material and Methods” have been removed, and the rest of the headings have been updated with correct numbers and fonts.
  • A schematic (figure 3) has been added showing some of the possible effects with L-carnitine on page 10.
  • Table has been adjusted because of the movement of the words.

Reviewer 2 Report

This review manuscript entitled “The Importance of the Fatty Acid Transporter L-carnitine, in Non-Alcoholic Fatty Liver Disease  (NAFLD)” led by Savic D et al., discuss a review of literature on the importance of L-Carnitine in fatty liver disease in particular with primary carnitine deficiency and the beneficial role of carnitine supplementation. The following comments were written for the improvement of this comprehensive review articles and are ordered chronological with page number.

Comments:

  • Carnitine absorption is 5-25% if supplemented orally, this limits the oral supplementation can be discussed further
  • Page 2, line 80 and line 83 needs a citation
  • Carnitine is synthesized from protein degradation as depicted in figure 1; can autophagy increase the synthesis of carnitine for its protective role?
  • Figure 1 is missing the details of 3-hydroxy-N-trimethyllysine aldolase (HTMLA), and 4-N-trimethylaminobutyraldehyde dehydrogenase (TMABA-DH). Also where does these reactions happen lysosomes, mitochondria or cytosol? Consider including the carnitine transporters in this figure.
  • Page 3, line 101 and line 104 needs a citation
  • What was the concentration of free carnitine in the inner membrane space for the activity of CACT?
  • Page 4 line 121, remove ‘$’ symbols
  • Consider including abbreviations section, e.g. TAGS is triglycerides should be expanded atleast once
  • Authors should consider providing comprehensive literature of primary carnitine deficiencies. Consider including the incidence, SLC22a5 gene mutations and these patients are managed and development of fatty liver disease. These information would be great addition to this review article
  • Page 6 line 213 needs a citation
  • Page 6 typo “hepatocellular”
  • Please double check all the references, e.g. Ref 76 does not have any measurement of carnitine as claimed in this manuscript
  • Authors should also consider adding the disadvantage of carnitine supplementation or gut microbial derived carnitine metabolism and generation of trimethyl amine as described in 23563705
  • Consider citing the updated article on Randle cycle, PMID: 19531645
  • Also consider adding a future directions in carnitine supplementation and its protective role for NAFLD.

Author Response

Dear Reviewer

Thank you for your feedback and for making this manuscript even better. We have addressed all the comments as seen below.

  • Page 2, line 189-198, the carnitine absorption have been discussed a bit further. These low limits have been shown in healthy humans, however, we yet have to investigate the absorption rates in diseased patients.
  • Line 204, 208: We have added the reference to “figure 1” and “figure 2”.
  • Line 1448. It is unclear in the literature if autophagy can increase the synthesis of carnitine, but carnitine can elevate autophagy in the skeletal muscle. A paper showed that L-carnitine treatment induced autophagy through PPARgamma, and thereby improved insulin sensitivity, this has been added.
  • Figure 1. We wanted to simplify the figure why the additional reactions have been written in the figure text and there are no carnitine transporters. That the additional steps are missing in the figure, but are explained in the figure text has been made clearer.
  • Line 205-206: The place of the reactions have been written in the text, all happen in the cytosol, except TMLD that has been shown to be localized in the mitochondrial matrix.
  • Line 333, 335: We have added the reference to the figures.
  • Not sure about the concentration in the inner membrane space.
  • Line 426: $ in the word “beta-oxidation” have been removed
  • Line 586: TAGS have been expanded to triglycerides.
  • Line 561-577. More information has been added about PCD, but since there is limited detailed investigation about hepatomegaly in PCD, due to death, we have not been able to include more information.
  • Line 886: The citations were added to Figure 1.
  • Line 902: “hepatcolellular” has been edited to “hepatocellular”
  • Line 920-927: References have been double-checked. Reference 76 has been updated.
  • Line 1146-1158. The paper from Koeth et all has been referenced. This paper shows that the detrimental effect is observed through consumption of meat and not pure oral L-carnitine, the difference is still unclear, but papers have shown that uptake in the intestines is much higher through food, than pure oral L-carnitine.
  • Line 1431: We have referenced the updated Randle cycle.
  • Line 1592-1598: In the conclusion we have added a bit about future investigation of L-carnitine
